# Silicon Wafer CMP Slurry Using a Hydrolysis Reaction Accelerator with an Amine Functional Group Remarkably Enhances Polishing Rate

**DOI:** 10.3390/nano12213893

**Published:** 2022-11-04

**Authors:** Jae-Young Bae, Man-Hyup Han, Seung-Jae Lee, Eun-Seong Kim, Kyungsik Lee, Gon-sub Lee, Jin-Hyung Park, Jea-Gun Park

**Affiliations:** 1Department of Energy Engineering, Hanyang University, Seoul 04763, Korea; 2Department of Nanoscale Semiconductor Engineering, Hanyang University, Seoul 04763, Korea; 3Department of Electronic Engineering, Hanyang University, Seoul 04763, Korea; 4UB Materials Inc., Yongin 17162, Korea

**Keywords:** colloidal silica, chemical–mechanical planarization, silicon wafer (Si wafer), hydrolysis reaction accelerator, processing in memory (PIM), 3D heterogeneous packaging

## Abstract

Recently, as an alternative solution for overcoming the scaling-down limitations of logic devices with design length of less than 3 nm and enhancing DRAM operation performance, 3D heterogeneous packaging technology has been intensively researched, essentially requiring Si wafer polishing at a very high Si polishing rate (500 nm/min) by accelerating the degree of the hydrolysis reaction (i.e., Si-O-H) on the polished Si wafer surface during CMP. Unlike conventional hydrolysis reaction accelerators (i.e., sodium hydroxide and potassium hydroxide), a novel hydrolysis reaction accelerator with amine functional groups (i.e., 552.8 nm/min for ethylenediamine) surprisingly presented an Si wafer polishing rate >3 times higher than that of conventional hydrolysis reaction accelerators (177.1 nm/min for sodium hydroxide). This remarkable enhancement of the Si wafer polishing rate for ethylenediamine was principally the result of (i) the increased hydrolysis reaction, (ii) the enhanced degree of adsorption of the CMP slurry on the polished Si wafer surface during CMP, and (iii) the decreased electrostatic repulsive force between colloidal silica abrasives and the Si wafer surface. A higher ethylenediamine concentration in the Si wafer CMP slurry led to a higher extent of hydrolysis reaction and degree of adsorption for the slurry and a lower electrostatic repulsive force; thus, a higher ethylenediamine concentration resulted in a higher Si wafer polishing rate. With the aim of achieving further improvements to the Si wafer polishing rates using Si wafer CMP slurry including ethylenediamine, the Si wafer polishing rate increased remarkably and root-squarely with the increasing ethylenediamine concentration.

## 1. Introduction

Recently, processing in memory (PIM), performing computing within memory, has been researched intensively with the aim of accelerating artificial intelligence systems. It is able to considerably enhance the training and inference speed of artificial intelligence and notably reduce power consumption by decreasing the usage frequency of data bus and CPU and memory [1,2,3,4,5]. Specifically, high-bandwidth memory (HBM)-PIM is able to achieve a computing performance enhancement of ~2 times, while obtaining a reduction in power consumption of ~70% and requires the bonding of multiple very thin memory chips through silicon via (TSV) [6]. These multiple thin memory chips can be fabricated by grinding followed by CMP of the Si wafers implemented in the memory cells [7,8,9,10,11]. In addition, three-dimensional (3D) heterogeneous packages of logic and memory chips have been studied as an alternative solution to overcome the physical limitations of scaling logic devices down to dimensions of less than 2 nm, which requires thin logic and memory chips [12,13,14,15,16,17]. Thinner logic and memory chips have been produced by grinding followed by CMP of the Si wafers installed in logic and memory devices. Thus, the CMP of the Si wafers can be employed as an essential process in the HBM-PIM and 3D heterogeneous packages of logic and memory chips. The CMP of Si wafers contributes to the elimination of grinding damage and a remarkable reduction in the surface roughness of the thin Si wafer, essentially achieving a remarkably high Si wafer polishing rate (e.g., >500 nm/min) and providing an Si wafer with relatively low surface roughness (e.g., <1 nm for a scanning area of 5 × 5 μm^2^).

In general, the CMP of Si wafer surfaces is performed by accelerating the hydrolysis reaction to form Si(OH)_x_ bonds on the polished Si wafer surface during CMP using an Si wafer CMP slurry that includes colloidal silica abrasives and a hydrolysis reaction accelerator, whereby a greater extent of the hydrolysis reaction on the polished Si wafer surface results in a higher Si wafer polishing rate [18,19,20,21,22,23,24,25,26]. Conventionally, the hydrolysis reaction acceleration is produced by a chemical reaction between OH^−^ in the CMP slurry and the Si atoms on the polished Si wafer surface; thus, NaOH or KOH have typically been used. However, CMP slurry for Si wafers using NaOH and KOH present notably low Si wafer polishing rates (177.1 nm/min for NaOH at a pH of 10.90, and 193.2 nm/min for KOH at a pH of 10.90), as shown in Appendix A. In particular, the evidence of a hydrolysis reaction on the Si wafer surface via a chemical reaction between covalent bonded Si atoms and OH^−^ ions has not been evidently proven by investigating the chemical compositions of the polished Si wafer surface at CMP [18]. Moreover, commercial Si wafer CMP slurries demonstrate an Si wafer polishing rate of 316.7~372.3 nm/min, as shown in Appendix A. Furthermore, it has been reported that the Si wafer CMP slurry using a hydrolysis reaction accelerator with amine functional group has presented an insufficient Si wafer polishing rate even though using a relatively hard CMP pad, as shown in Appendix A [27,28]. In particular, the CMP mechanism by which the hydrolysis reaction accelerators, i.e., NaOH, KOH, or polyamine having amine functional groups, could enhance considerably have not been clearly demonstrated [29,30].

Thus, in this study, a novel hydrolysis reaction accelerator possessing amine functional groups in the Si wafer CMP slurry (i.e., ethylenediamine: EDA) was designed to remarkably enhance the Si wafer polishing rate compared to NaOH and KOH (i.e., ~552.8 nm/min at a pH of 10.90). Specifically, the Si wafer polishing rate increased logarithmically with the concentration of the hydrolysis reaction accelerator with amine functional group up to a rate of ~872 nm/min, as shown in Appendix A. In addition, the mechanism by which the EDA notably enhanced the Si wafer polishing rate was determined by reviewing chemical properties such as the hydrolysis reaction (i.e., Si(OH)_x_ bond intensity) and the degree of adsorption of the Si wafer CMP slurry using EDA, as well as mechanical properties such as the electrostatic repulsive force.

## 2. Materials and Methods

### 2.1. Materials

The CMP experiments were conducted on commercially available (1 0 0) silicon wafers. In this experiment, 83.3 nm colloidal silica abrasives were synthesized with 60-nm-diameter colloidal silica dispersed in deionized (DI) water. The Si wafer CMP slurries were composed of the colloidal silica abrasives of 1 wt% and the hydrolysis reaction accelerator of 0~0.125 wt% for NaOH (Sigma Aldrich, ACS reagent, ≥97.0%, pellets), 0~0.069 wt% for KOH (Sigma Aldrich, ACS reagent, ≥85.0%, pellets), and 0~0.10 wt% for EDA (Sigma Aldrich, ReagentPlus^®^, ≥99%), DETA (Sigma Aldrich, ReagentPlus^®^, ≥99%), and TETA (Sigma Aldrich, technical grade, 60%), respectively. DETA and TETA refer to diethylenetriamine and triethylenetetramine, respectively. Please note that the pH of the Si wafer CMP slurry containing only colloidal silica abrasives was 9.70, and the pH of the Si wafer CMP slurries using various hydrolysis reaction accelerators increased with the concentration of the hydrolysis reaction accelerators. In addition, particularly, the secondary colloidal silica abrasive sizes of the Si wafer CMP slurries using EDA concentration from 0 to 0.10 wt% were observed at 80~85 nm, as shown in Appendix A.

### 2.2. CMP Conditions

Eight-inch Si wafers with (1 0 0) orientations were cut into 4 × 4 cm squares. The CMP process was carried out on the Si wafer surface using a CMP polisher (POLI-300, G&P Tech. Inc., Busan, Korea) attached with a rectangular-grooved CMP pad (SUBA 600, Nitta Haas Inc., Osaka, Japan). The polishing pad was conditioned with DI water and a diamond disk conditioner for 30 min before polishing, and then pre-polishing was performed on three dummy wafers prior to the actual polishing of the Si wafer surface. Pad conditioning and cleaning were carried out after each polishing stage for the various slurries. The applied head pressure was 5.7 psi, the rotation speed of the plate with the CMP pad attached was 69 rpm, and the rotation speed of the carrier holding the Si wafer samples was 71 rpm. In all CMP experiments, the flow rate of the CMP slurry was fixed at 100 mL/min, and each polishing stage was set to 5 min. After CMP for 5 min, all Si wafer samples were buffed with DI water for 30 s to remove abrasives and debris remaining on the Si wafer surface.

### 2.3. Characterization

The polishing rate of the Si wafer was estimated by measuring the mass before and after CMP (i.e., $Polishing\;rate_{Si}=\Delta m/(\rho \cdot A\cdot t$)); specifically, the Si wafer polishing rate can be calculated by weight change (Δm), where m, ρ, A, and t are the weight of the Si wafer, the density of Si, the area of the polished Si wafer surface, and polishing time, respectively, using scales (Balance XS205DU, Mettler-Toledo International Inc., Greifensee, Switzerland). The SEM images of the colloidal silica abrasives were obtained using field emission scanning electron microscopy (FE-SEM, S-4800, Hitach High-Tech Co., Tokyo, Japan) with an accelerating voltage of 15 kV; i.e., the diameter of colloidal silica abrasive was 60 nm. The chemical composition of the polished Si wafer surface was characterized using XPS (X-ray photoelectron spectroscopy; K-Alpha+, Thermo Fisher Scientific Co., Inc., Waltham, MA, USA) at 12 keV and 6 mA with A1Kα (1486.6 eV). The contact angles were measured using a contact angle meter (GBX Instrument, DIGIDROP, Dublin, Ireland) by dropping 0.01 mL of the Si wafer CMP slurries on the polished Si wafer surface. The secondary colloidal silica abrasive size, the zeta-potential of the colloidal silica abrasives in the CMP slurry, and the zeta-potential of the polished Si wafer after CMP were analyzed using a particle analyzer (ELSZ2+, Otsuka Electronics Co., Inc., Osaka, Japan). The surface roughness (i.e., the average root mean square (RMS) roughness) of the polished Si wafer was estimated by atomic force microscopy (AFM, Park system, Suwon, Korea) with a 5 μm × 5 μm scanning area.

## 3. Results and Discussion

The mechanism of CMP is based on (i) the chemical reaction between the chemicals in the slurry (i.e., dispersant, titrant, polishing rate accelerator or inhibitor, etc.) and the film surface being polished, (ii) the degree of adsorption of the slurry on the film surface being polished, (iii) the mechanical rubbing between the charged abrasives and the chemically reacted film surface being polished, and (iv) the coming in- or out-charge abrasives and CMP debris produced during CMP, as governed by Stoke’s law [31,32,33,34,35,36,37,38,39,40,41,42,43,44,45,46]. Thus, the metrics of CMP performance (i.e., film polishing rate, film polishing rate selectivity, remaining abrasives and debris, CMP-induced scratches, dishing, and erosion) are principally determined by the CMP mechanism, including (i)–(iv) described above. To understand how our novel Si wafer CMP slurry so remarkably enhanced the Si wafer polishing rate, the CMP mechanism was reviewed on the basis of the correlation between CMP performance and (i)–(iv), described above.

### 3.1. Dependency of Silicon Wafer Polishing Rate on the Chemical Structure of the Hydrolysis Reaction Accelerator (i.e., NaOH, KOH, EDA, DETA, TETA) with Amine Functional Groups and pH

To enhance the Si wafer polishing rate, it is necessary to produce a hydrolysis reaction on the Si wafer surface between the Si atoms and hydroxide (i.e., OH^−^) to form Si(OH)_x_. Five different types of chemicals were tested as hydrolysis reaction accelerators with respect to their impact on the Si wafer polishing rate; i.e., NaOH, KOH, EDA, DETA, and TETA. NaOH and KOH are conventional hydrolysis reaction accelerators (referred to as a pH titrant), while EDA, DETA, and TETA, as novel hydrolysis reaction accelerators, include amine functional groups. The Si wafer polishing rate was estimated after performing CMP for 5 min using CMP slurries including 60-nm-diameter colloidal silica abrasives of 1 wt% and the five different hydrolysis reaction accelerators. The Si wafer polishing rate increased linearly from 139.5 to 177.1 nm/min and 139.5 to 193.2 nm/min, respectively, when increasing the NaOH concentration from 0 to 0.125 wt% and increasing the KOH concentration from 0 to 0.069 wt%, respectively, as shown in Figure 1a. Otherwise, for EDA, the Si wafer polishing rate increased root-squarely from 139.5 to 552.8 nm/min, for DETA, the Si wafer polishing rate increased root-squarely from 139.5 to 617.2 nm/min, and for TETA, the Si wafer polishing rate increased rapidly from 139.5 to 526.0 nm/min, becoming saturated at ~499.1 nm/min when increasing the concentration of the hydrolysis reaction accelerator from 0 to 0.10 wt%. The Si wafer polishing rate of EDA including an amine functional group (i.e., 552.8 nm/min at 0.10 wt%) was ~3 times higher than that of NaOH (i.e., 177.1 nm/min at 0.125 wt%) and KOH (i.e., 193.2 nm/min at 0.069 wt%), which is analyzed in further detail later. The highest values of Si wafer polishing rate were achieved by DETA (i.e., 617.2 nm/min at 0.10 wt%), EDA (i.e., 552.8 nm/min at 0.10 wt%), and TETA (i.e., 499.1 nm/min at 0.10 wt%). Since the Si wafer polishing rate is fundamentally proportional to the extent of the hydrolysis reaction on the Si wafer during CMP, the concentration of OH^−^ was calculated by measuring the pH of the CMP slurries that included a hydrolysis reaction accelerator as function of pH, as shown in Figure 1b. When the concentration of the hydrolysis reaction accelerator was increased from 0 to 0.125 wt% for NaOH, 0 to 0.069 wt% for KOH, and 0 to 0.10 wt% for EDA, DETA, and TETA, the pH increased from 9.7 to 10.90 for NaOH, from 9.7 to 10.90 for KOH, from 9.7 to 10.90 for EDA, from 9.7 to 10.81 for DETA, and from 9.7 to 10.85 for TETA. All hydrolysis reaction accelerators exhibited dependencies of pH on the concentration of the hydrolysis reaction accelerators that were almost same, indicating that the increase in OH^−^ concentration would be almost the same for all hydrolysis reaction accelerators when increasing the hydrolysis reaction accelerator concentration from 0 to 0.125 wt% for NaOH, 0 to 0.069 wt% for KOH, and 0 to 0.10 wt% for EDA, DETA, and TETA. However, the dependencies of the Si wafer polishing rate on the hydrolysis reaction accelerator concentration were completely different between the conventional hydrolysis reaction accelerators (i.e., NaOH and KOH) and the hydrolysis reaction accelerators that included amine functional groups (i.e., EDA, DETA, and TETA), as shown in Figure 1b. These results imply that the Si wafer polishing rate is not simply determined by the OH^−^ concentration generated by the hydrolysis reaction accelerator. Thus, the mechanism by why hydrolysis reaction accelerators that include amine functional groups are able to present such a remarkable Si wafer polishing rate was characterized by reviewing (i) the extent of hydrolysis reaction, as a chemical property, (ii) the degree of adsorption of the CMP slurry on the polished Si wafer surface, as another chemical property, and (iii) the electrostatic repulsive force between colloidal silica abrasives and the polished Si wafer surface as a mechanical property. In particular, it was confirmed that the dependencies of the Si wafer polishing rate on the CMP head and CMP platen rotation speeds as well as polishing time presented a typical Preston behavior, indicating that the Si wafer CMP slurry including a hydrolysis reaction accelerator with amine functional group would prefer to be a mechanically dominant CMP, as shown in Appendix A. Note that the Si wafer polishing rate was determined on the basis of both chemical and mechanical properties [18,19,20,22,26,47,48,49,50,51,52]. In addition, the Si wafer CMP slurry using the hydrolysis reaction accelerator with amine functional group (i.e., EDA) evidently presented a remarkably high Si wafer polishing rate at a relatively low rotation speed (i.e., 70 rpm), a low head pressure (i.e., 4 psi), and a softer pad, as shown in Appendix A.

### 3.2. Dependencies of Chemical Properties (i.e., Chemical Composition and the Degree of Adsorption via Hydrolysis Reactions on the Concentration of the Hydrolysis Reaction Accelerator EDA) in the Si Wafer CMP Slurry

EDA in particular was chosen to determine why the hydrolysis reaction accelerators that included amine functional groups showed such a notable Si wafer polishing rate compared to conventional hydrolysis reaction accelerators (i.e., NaOH and KOH) due to its relatively low production price compared to the other hydrolysis reaction accelerators containing amine functional groups, as shown in Appendix A. The EDA (C_2_H_4_(NH_2_)_2_) was dissociated with positively charged C_2_H_4_{(NH_3_)^+^}_2_ and negatively charged OH^−^ in the CMP slurry including 60-nm-diameter colloidal silica abrasive and DIW at pH 9.7, as shown in Equation (1). Thus, the pH of the CMP slurry increased logarithmically with increasing EDA concentration, as shown in Figure 1b. In addition, the generated OH^−^ was chemically reacted with the Si wafer surface, producing Si(OH)_2_^2+^ and 2e^−^, as shown in Equation (2). Eventually, Si(OH)_2_^2+^ was chemically reacted with 2OH^−^, producing a hydrolysis-reacted Si wafer surface [Si(OH)_4_] and 2e^−^, as shown in Equation (3) [18].
C_2_H_4_(NH_2_)_2_ + 2H_2_O ⟶ C_2_H_4_{(NH_3_)^+^}_2_ + 2OH^−^(1)
Si + 2OH^−^ ⟶ Si(OH)_2_^2+^ + 2e^−^(2)
Si(OH)_2_^2+^ + 2OH^−^ ⟶ Si(OH)_4_ + 2e^−^(3)

To estimate the extent of the hydrolysis reaction as a function of EDA concentration for the polished Si wafer using the CMP slurry, the Si-O-H bond spectral intensity was characterized using XPS. The Si-O-H and Si-O-Si bonds in the O 1s spectra were found at 533.2 and 532.7 eV, respectively, as shown in Figure 2a. When the EDA concentration was increased from 0 to 0.10 wt%, the normalized percentage of Si-O-H at 533.2 eV in the XPS spectra increased almost linearly from 18.526 to 42.184%, while that of Si-O-Si at 532.7 eV decreased almost linearly from 81.474 to 57.816%, as shown in Figure 2b. These results mean that the addition of EDA in the CMP slurry accelerates the hydrolysis chemical reaction on the Si wafer surface rather than chemical oxidation occurring during CMP. In addition, since the OH^−^ concentration increased exponentially with increasing EDA concentration, as shown in Figure 1b, the extent of the hydrolysis chemical reaction (i.e., normalized Si-O-H percentage) increased linearly with increasing EDA concentration, as shown in Figure 2b. The SiO_2_ at 2p and Si at 2p^1/2^ and 2p^2/3^ bond spectra were found at 103.5, 100.1 eV and 99.5 eV, respectively, as shown in Figure 2c. Their normalized XPS spectra were independent of the concentration of hydrolysis reaction accelerator, again meaning that the hydrolysis reaction for forming Si-O-H is dominant, rather than chemical oxidation occurring during CMP, as shown in Figure 2d. However, the extent of the hydrolysis chemical reaction itself does not explain why the Si wafer polishing rate with EDA was remarkably higher than that with NaOH and KOH, since the OH^−^ concentrations with EDA, NaOH, and KOH were simply proportional to the pH of the slurries, as shown in Figure 1b. Thus, another chemical property of the Si wafer polishing slurry, i.e., the degree of adsorption of the slurry on the polished wafer, was investigated as a function of the chemical structure of the hydrolysis reaction accelerator and its concentration.

The degree of adsorption (i.e., the contact angle) was estimated by dropping 0.01 mL of slurry including specific concentrations of hydrolysis reaction accelerator onto the polished Si wafer surface. The contact angles for NaOH and KOH decreased considerably and almost linearly from 50.5 to 31.83° for NaOH and 50.5 to 30.08° for KOH, while the NaOH and KOH concentration in the Si wafer CMP slurries increased from 0 to 0.125 wt% and 0 to 0.069 wt%, respectively, as shown in Figure 3a. The addition of NaOH or KOH to the Si wafer CMP slurry enhanced the OH^−^ concentration in the slurry, i.e., the OH^−^ concentration increased exponentially with increasing NaOH or KOH concentration, as shown in Figure 1b. As confirmed by XPS, since the Si-O-H spectral peak intensity increased with increasing OH^−^ concentration in the CMP slurry, the extent of the hydrolysis chemical reaction on the polished Si wafer surface directly influenced the contact angle of the CMP slurry, i.e., a higher OH^−^ concentration in the CMP slurry tended to result in a higher extent of hydrolysis chemical reaction on the polished Si wafer surface. As a result, the contact angle of the CMP slurry including NaOH or KOH decreased with increasing OH^−^ concentration in the CMP slurry. As another reason for the decrease in contact angle with increasing OH^−^ concentration, the dependencies of the zeta-potentials of the colloidal silica abrasives and the polished Si wafer surface on the OH^−^ concentration were measured as a function of the NaOH or KOH concentration. The zeta-potentials of the colloidal silica abrasives increased slightly from −44.90 to −46.06 mV, while that of the polished Si wafer surface was almost independent of NaOH or KOH concentration, as shown in Figure 4a,b. These results indicate that the decrease in the contact angle in the CMP slurry resulting from the increase in NaOH or KOH concentration was mainly related to the increase in the extent of the hydrolysis reaction between OH^−^ and the Si wafer surface rather than the change in the zeta-potential of the colloidal silica abrasives and the polished Si wafer surface.

Otherwise, the contact angle for EDA for 0.02 wt% dropped abruptly from 50.5 to 24.18°, and then noticeably decreased from 24.18 to 14.55° with further increases in EDA concentration in the Si wafer CMP slurries. The contact angle for EDA at 0.10 wt% (i.e., 14.55°) was ~2 times lower than those for NaOH at 0.125 wt% (i.e., 31.83°) and KOH at 0.069 wt% (i.e., 30.08°). Since the OH^−^ concentrations generated by the hydrolysis reaction accelerators in the Si wafer CMP slurry were almost same among NaOH, KOH, and EDA, as determined by the pH of the CMP slurry, as shown in Figure 1b, the difference in the contact angle between EDA, NaOH, and KOH at the same pH must be caused by a different chemical property of the CMP slurry, rather than the difference in the extent (i.e., OH^−^ concentration) of the hydrolysis chemical reaction on the polished Si wafer surface during CMP. This other chemical property of the CMP slurry must be associated with the surface charge (i.e., zeta-potential) of the colloidal silica abrasives and the polished Si wafer surface affected by the dissociated hydrolysis reaction accelerator in the Si wafer CMP slurry. As shown in Equation (1), EDA was dissociated from the positively charged C_2_H_4_{(NH_3_)^+^}_2_ and the negatively charged 2OH^−^. The negatively charged OH^−^ in the CMP slurry was employed to accelerate the hydrolysis chemical reaction between OH^−^ and the Si wafer surface during CMP. Additionally, the positively charged C_2_H_4_{(NH_3_)^+^}_2_ in the CMP slurry tended to be adsorbed onto the strongly negatively charged colloidal silica abrasives and the negatively charged hydrolysis-reacted Si wafer surface during CMP. Thus, the zeta-potential of the colloidal silica abrasives decreased slightly from −44.90 to −43.53 mV when the EDA concentration was increased to 0.06 wt% and then abruptly decreased from −43.53 to −36.78 mV with further increases in EDA concentration, evidencing the adsorption of the positively charged C_2_H_4_{(NH_3_)^+^}_2_ on the strongly negatively charged colloidal silica abrasive, as shown in Figure 4a. In addition, the zeta-potential of the hydrolysis-reacted polished Si wafer surface decreased linearly and noticeably from −33.48 to −28.37 mV when the EDA concentration was increased from 0 to 0.10 wt%, evidencing the adsorption of the positively charged C_2_H_4_{(NH_3_)^+^}_2_ onto the strongly negatively charged polished Si wafer surface, as shown in Figure 4b. Thus, the decrease in the zeta-potential of both the colloidal silica abrasives and the polished Si wafer surface with the addition of EDA resulted in a decrease in the contact angle of the CMP slurry since the decrease in the electrostatic repulsive force between colloidal silica abrasives and the polished Si wafer surface could enhance the contact angle of the CMP slurry on the polished Si wafer surface. Therefore, the dependency of the contact angle of the CMP slurry on EDA concentration was principally associated with the OH^−^ concentration increase as well as the decrease in the electrostatic repulsive force between colloidal silica abrasives and the polished Si wafer surface during CMP; i.e., higher OH^−^ concentration and lower electrostatic repulsive force led to the CMP slurry having a smaller contact angle. Moreover, since smaller contact angles of the CMP slurry on the polished Si wafer surface provide a higher probability of chemical and abrasive adsorption on the polished Si wafer surface, smaller contact angles result in a higher Si wafer polishing rate [37,53,54]. Thus, since the contact angle of the CMP slurry using EDA (i.e., 14.55° at 0.10 wt%) was ~2 times smaller than that using NaOH (i.e., 31.83° at 0.125 wt%) or KOH (i.e., 30.08° at 0.069 wt%), the Si wafer polishing rate for EDA (i.e., 552.8 nm/min at 0.10 wt%) was ~3 times higher than that for NaOH (i.e., 177.1 nm/min at 0.125 wt%) or KOH (i.e., 193.2 nm/min at 0.069 wt%), where other properties such as mechanical properties are present. These results imply that another chemical property (i.e., the degree of adsorption of the CMP slurry on the polished Si wafer surface) affects the Si wafer polishing rate.

### 3.3. Dependency of the Mechanical Properties (i.e., Electrostatic Force between Colloidal Silica Abrasives and the Si Wafer Surface) on the Hydrolysis Reaction Accelerator (NaOH, KOH, or EDA) Concentration

In general, the film polishing rate is principally determined by the chemical properties (i.e., hydrolysis reaction degree and the contact angle of the CMP slurry on the polished Si wafer surface) as well as the mechanical properties (i.e., electrostatic force between the colloidal silica abrasives and the polished Si wafer surface). Note that the film polishing rate is governed by Preston’ Equation (i.e., −dH/dt=c·P·v); in particular, the electrostatic force strongly affects a constant (c), where t, H, c, P, and v are time, film topography height, the material constant, CMP head pressure, and relative table velocity, respectively [55,56,57,58]. Thus, the relative electrostatic force between the colloidal silica abrasives and the polished Si wafer surface during CMP was calculated as a function of the concentration of the hydrolysis reaction accelerator. For the slurries using NaOH or KOH, it increased very slightly with increasing NaOH or KOH concentration, indicating that the electrostatic force for the slurries using NaOH or KOH was almost independent of the NaOH or KOH concentration, as shown in Figure 4c. Otherwise, the relative electrostatic repulsive force between the colloidal silica abrasives and the polished Si wafer surface during CMP using EDA remarkably and linearly decreased from 1503 to 1043 abs. since the dissociated positively charged C_2_H_4_{(NH_3_)^+^}_2_ was adsorbed on both negatively charged colloidal silica abrasives and the polished Si wafer surface; i.e., higher EDA concentrations in the CMP slurry resulted in lower relative electrostatic repulsive force between the colloidal silica abrasives and the polished Si wafer surface. Note that the colloidal silica abrasives at the CMP slurry of pH 9.7 themselves presented a highly negatively charged zeta-potential of −44.9 mV, as shown in Figure 4a. The addition of the EDA having a positively charged amine functional group into the CMP slurry including colloidal silica abrasives at pH 9.7 modified to reduce remarkably the zeta-potential of the colloidal silica abrasives via coating EDA on the negatively charged colloidal silica abrasives, i.e., from −44.9 to −36.78 mV, when the EDA concentration increased from 0 to 0.1 wt% [54]. It was also modified to decrease considerably the zeta-potential of the polished Si wafer surface via coating EDA on the negatively charged the polished Si wafer surface, i.e., from −33.48 to −28.37 mV, when the EDA concentration increased from 0 to 0.1 wt%, as shown in Figure 4b [37]. As a result, the zeta-potential decrease in both the colloidal silica abrasives and the polished Si wafer surface due to coating EDA on them reduced significantly the repulsive force between the colloidal silica abrasives and the polished Si wafer surface via Coulombic interaction between abrasives and film surface (i.e., $F=q_{1}\cdot q_{2}/r^{2}$, where F, $q_{1}$, $q_{2}$, and r are the repulsive force between the colloidal silica abrasives and the polished Si wafer surface, the zeta-potential of the colloidal silica abrasives, the zeta-potential of the polished Si wafer, and distance between the colloidal silica abrasives and the polished Si wafer surface, respectively), as shown in Figure 4c [46,47,48,49,50]. To understand how the relative electrostatic repulsive force influences the Si wafer polishing rate, the correlation was determined between the Si wafer polishing rate and relative electrostatic repulsive force for NaOH, KOH, and EDA, as shown in Figure 4d. For all of the hydrolysis reaction accelerators, the Si wafer polishing rate decreased notably, from 552.8 to 139.5 nm/min, when the relative electrostatic repulsive force was increased from 1043 to 1503 abs., i.e., higher relative electrostatic repulsive force led to lower Si wafer polishing rate [48,49,50,51,52]. In particular, the relative electrostatic repulsive force (i.e., 1503 abs.) for NaOH and KOH was independent of the NaOH or KOH concentration, while for EDA, it decreased with increasing EDA concentration (i.e., from 1503 to 1043 abs.). Thus, the Si wafer polishing rate increased remarkably with increasing EDA concentration, but it did not change with increasing NaOH or KOH concentration. These results imply that the difference in the Si wafer polishing rate between NaOH, KOH, and EDA was evidently caused by the chemical properties (i.e., the extent of the hydrolysis reaction and the degree of adsorption on the Si wafer surface of the CMP slurry including EDA) as well as the mechanical properties (i.e., the relative electrostatic repulsive force between the colloidal silica abrasives and the polished Si wafer surface).

## 4. Conclusions

Unlike conventional hydrolysis reaction accelerators (i.e., NaOH and KOH), our novel hydrolysis reaction accelerator including amine functional groups (i.e., EDA) achieved a remarkably high Si wafer polishing rate of 552.8 nm/min at 0.10 wt%, which is ~3 times higher than the Si wafer polishing rate achieved when using NaOH at 0.125 wt% or KOH at 0.069 wt%. In addition, the Si wafer CMP slurry using EDA at 0.10 wt% accomplished a noticeably low surface roughness of 0.353 nm for a scanning area of 5 × 5 μm^2^ after CMP, as shown in Appendix A. The addition of EDA to the Si wafer CMP slurry through its dissociation from C_2_H_4_{(NH_3_)^+^}_2_ and OH^−^ notably enhanced (i) the hydrolysis reaction extent via the chemical reaction between the OH^−^ and Si atoms on the polished Si wafer surface and (ii) the adsorption degree of the Si wafer CMP slurry on the polished Si wafer surface by reducing the zeta-potentials of both the colloidal silica abrasives and the polished Si wafer surface, i.e., higher EDA concentrations in the Si wafer CMP slurry led to higher hydrolysis reaction extents and adsorption degree of the CMP slurry (i.e., a smaller contact angle of the CMP slurry on the polished Si wafer surface). This considerably decreased (iii) the relative electrostatic repulsive force between the colloidal silica abrasives and the polished Si wafer surface, i.e., higher EDA concentrations in the Si wafer CMP slurry led to lower relative electrostatic repulsive force between the colloidal silica abrasives and the polished Si wafer surface. Points (i) and (ii) describe improvements to the chemical properties, while point (iii) describes an improvement of the mechanical properties during Si wafer CMP. In general, because the higher extent of the hydrolysis reaction, higher CMP slurry adsorption, and lower relative electrostatic repulsive force lead to a higher Si wafer polishing rate during CMP, the Si wafer polishing rate when using the novel hydrolysis reaction accelerator including amine functional groups (i.e., EDA, DETA, and TETA) was remarkably enhanced compared to that when using conventional hydrolysis reaction accelerators (i.e., NaOH and KOH). In particular, a further increase in the EDA concentration in the Si wafer CMP slurry was able to increase the Si wafer polishing rate to above 872 nm/min, as shown in Appendix A. The addition of a hydrolysis reaction accelerator possessing amine functional groups (i.e., EDA, DETA, and TETA) in the Si_x_Ge_1−x_-film CMP slurry would remarkably enhance the polishing rate of the Si_x_Ge_1−x_-film surface during CMP, and further study on this is necessary.

## Figures and Tables

**Figure 1 nanomaterials-12-03893-f001:**
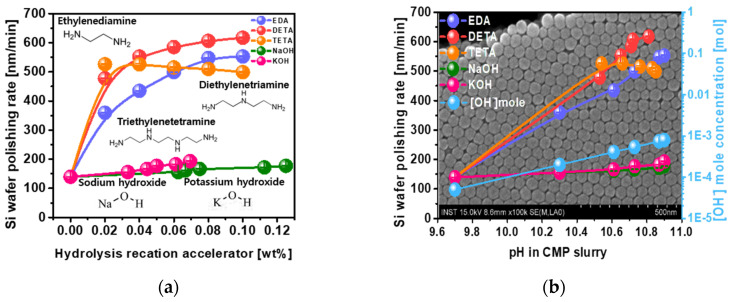
Dependencies of the Si wafer polishing rate on the chemical structure of a hydrolysis reaction accelerator and its concentration. (**a**) Effect of the hydrolysis reaction accelerator including amine functional group on the enhancement of Si wafer polishing rate and (**b**) correlation between the Si wafer polishing rate and OH^−^ concentration. The background Figure 1b is the SEM image of colloidal silica abrasives used in our study.

**Figure 2 nanomaterials-12-03893-f002:**
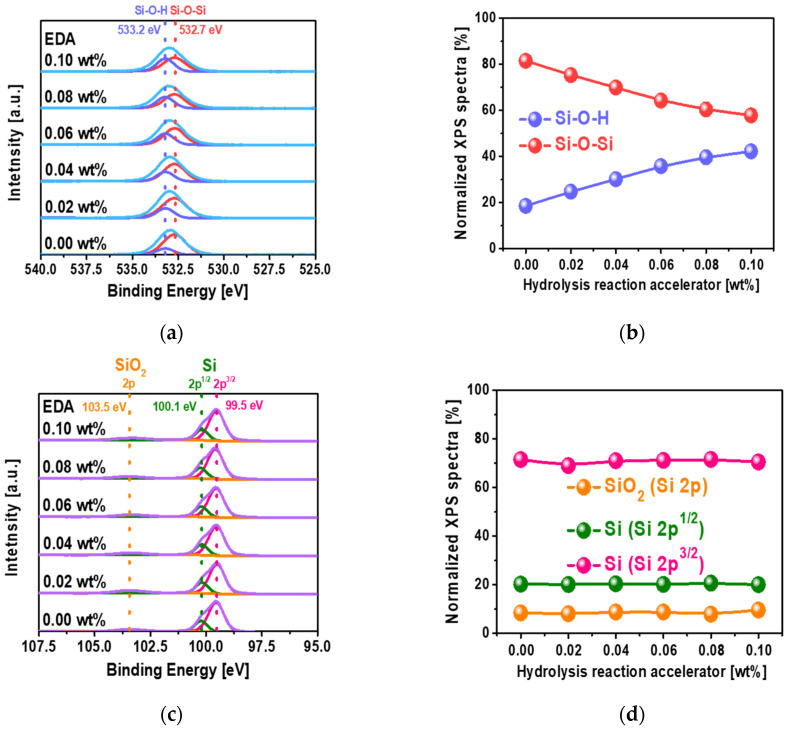
Dependency of the chemical compositions on the EDA concentration for the polished Si wafer surface after CMP, analyzed by XPS. (**a**) O 1s XPS spectra, (**b**) normalized XPS spectra percentage at O 1s XPS spectra depending on the EDA concentration, (**c**) Si 2p XPS spectra, and (**d**) normalized XPS spectra percentage at Si 2p XPS spectra depending on the EDA concentration.

**Figure 3 nanomaterials-12-03893-f003:**
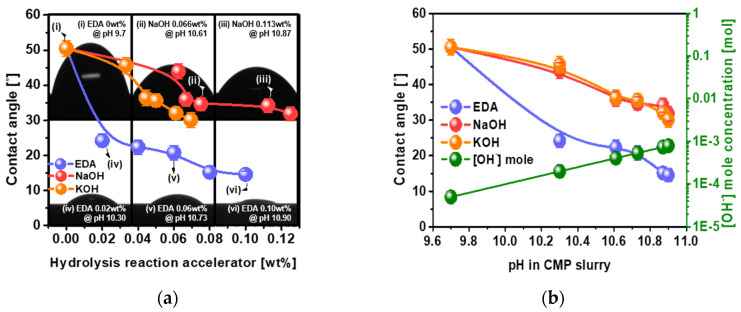
Dependencies of the degree of adsorption (i.e., contact angle) on the chemical structure of the hydrolysis reaction accelerators and their concentration at the polished Si wafer surface. (**a**) Contact angles of the CMP slurries using NaOH, KOH, and EDA on the polished Si wafer surface, depending on concentration. (**b**) Dependencies of contact angle on the pH of the CMP slurry using NaOH, KOH, or EDA.

**Figure 4 nanomaterials-12-03893-f004:**
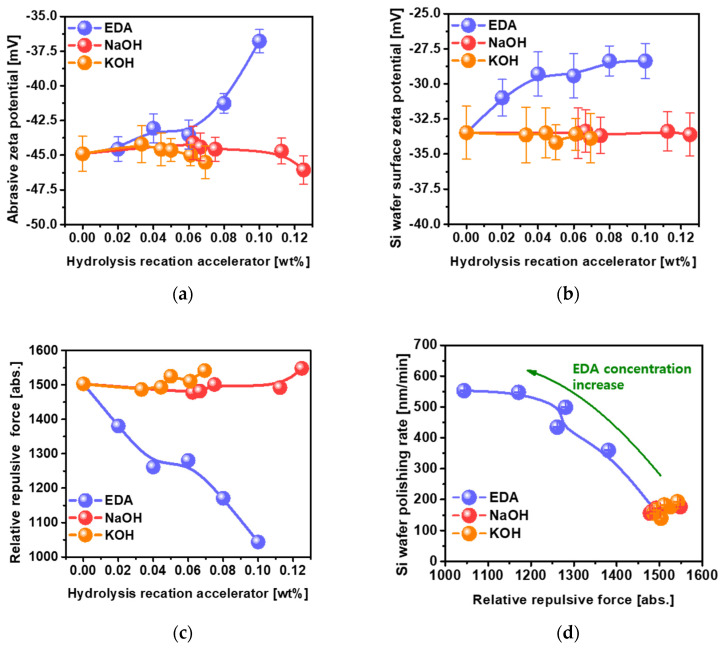
Influence of mechanical properties (i.e., relative electrostatic repulsive force between the colloidal silica abrasives and the polished Si wafer surface) on the Si wafer polishing rate. (**a**) Zeta-potentials of the colloidal silica abrasives in the Si wafer CMP slurries, (**b**) zeta-potentials of the polished Si wafer surface after CMP, (**c**) relative electrostatic repulsive force between the colloidal silica abrasives and the polished Si wafer surface, and (**d**) effect of the relative electrostatic repulsive force on the Si wafer polishing rate, depending on the chemical structure of the hydrolysis reaction accelerator (i.e., NaOH, KOH, and EDA) and their concentration.

## Data Availability

Data can be available upon request from the authors.

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
