# Peer review of "Silicon Wafer CMP Slurry Using a Hydrolysis Reaction Accelerator with an Amine Functional Group Remarkably Enhances Polishing Rate"

_nanomaterials, 2022, doi:10.3390/nano12213893_

Round 1
Reviewer 1 Report
1. The authors should compare with other studies that the polishing rate of wafers can be enhanced using hydrolysis reaction accelerators.
2. Authors should cite recent references to novel hydrolysis reaction accelerators with amine functional groups in the Introduction.
3. The authors should explain why they chose the composition of 1 wt% colloidal silica abrasive and 0 ~ 0.125 wt% NaOH (Sigma Aldrich), 0 ~ 0.069 wt% KOH and 0 ~ 0.10 wt% hydrolysis reaction accelerator For EDA, DETA and TETA respectively. The manufacturer and purity of these materials should also be described.
4. CMP rotation speed and polishing time will affect the polishing quality of Si wafers. The authors should investigate the effect of different rotational speeds on Si wafer polishing.
5. The description of Fig. 1 only illustrates the phenomenon of the silicon wafer polishing rate on the chemical concentration of the hydrolysis reaction accelerator. The authors should cite other studies or theories for a more in-depth study of why the chemical concentration of the hydrolysis reaction accelerator affects the polishing rate.
6. The description of the conclusion is too long, the conclusion should be briefly described.
7. In Figure 3 the authors should cite other studies or theories for a more in-depth study of why the contact angle depends on the chemical structure of the hydrolysis reaction accelerator and its concentration on the polished silicon wafer surface.
Reviewer 2 Report
Reviewer report on Manuscript Draft ‘Efficiency Enhancement in an Electrochemically Deposited Solar Cell developed using Kesterite Cu2ZnSnS4 via a P3HT:PCBM Polymer Layer’
In this research authors report a hydrolysis reaction accelerator possessing amine functional groups in the Si wafer CMP slurry. Ethylenediamine (EDA) was designed to remarkably enhance the Si wafer polishing rate compared to NaOH and KOH. Specifically, the Si wafer polishing rate increased logarithmically with the concentration of the hydrolysis reaction accelerator with amine functional group.
This research is technological, but not very innovative. Therefore, the manuscript is not recommended for publication.
Some specific comments:
Introduction is rather short and weak.
Investigations are very simple.
Statistics and validation of results is not well addressed in presented plots with data.
Reviewer 3 Report
The paper is interesting and well written. Here are my minor suggestions:
The abstract contains too many abbreviations e.g. the full name of the molecule ethylenediamine should be mentioned in the abstract.
In line 1 of page 2, (D) should be replaced by (3D).
In subsection 3.2, it is written that pH linearly depends on EDA concentration. pH likely varies linearly with log of EDA concentration.
In page 7, it is written "the increase in contact angle with increasing OH - concentration" while this angle decreases with OH- concentration as shown in figure 3b.
Measurements of contact angle show that EDA enhances the polishing paste adsorption. Repulse force calculation show that EDA reduces repulsion between abrasive and wafer. Could the authors provide a microscopic mechanism explaining why the zeta potential is modified by EDA and why this molecule enhances paste adsorption and reduces repulsion of the abrasive?
Round 2
Reviewer 1 Report
The author has revised the paper according to the comments of the reviewers. The structure of the paper and the description in English are clear, so the manuscript I recommend can be published.
Reviewer 2 Report
Reviewer report on Manuscript Draft ‘Efficiency Enhancement in an Electrochemically Deposited Solar Cell developed using Kesterite Cu2ZnSnS4 via a P3HT:PCBM Polymer Layer’
In this research authors report a hydrolysis reaction accelerator possessing amine functional groups in the Si wafer CMP slurry. Ethylenediamine (EDA) was designed to remarkably enhance the Si wafer polishing rate compared to NaOH and KOH. Specifically, the Si wafer polishing rate increased logarithmically with the concentration of the hydrolysis reaction accelerator with amine functional group.
Introduction was improved.
Some other parts of the manuscript were improved, therefore, manuscript eventually is suitable for publishing.